# Traumatic Events in Dual Disorders: Prevalence and Clinical Characteristics

**DOI:** 10.3390/jcm9082553

**Published:** 2020-08-06

**Authors:** Laura Blanco, Albert Sió, Bridget Hogg, Ricard Esteve, Joaquim Radua, Aleix Solanes, Itxaso Gardoki-Souto, Rosa Sauras, Adriana Farré, Claudio Castillo, Alicia Valiente-Gómez, Víctor Pérez, Marta Torrens, Benedikt L. Amann, Ana Moreno-Alcázar

**Affiliations:** 1Benito Menni Complex Assistencial en Salut Mental, Sant Boi de Llobregat, 08830 Barcelona, Spain; lbpresas@gmail.com (L.B.); asioeroles@gmail.com (A.S.); 2Department of Personality, Evaluation and Psychological Treatments, University of Barcelona, 08007 Barcelona, Spain; 3Centre Fòrum Research Unit, Institut de Neuropsiquiatria i Addiccions, Parc de Salut Mar, 08019 Barcelona, Spain; bridget.hogg@e-campus.uab.cat (B.H.); itxasogardoki@gmail.com (I.G.-S.); avaliente@parcdesalutmar.cat (A.V.-G.); amoreno.centreforum@gmail.com (A.M.-A.); 4Hospital del Mar Medical Research Institute (IMIM), 08003 Barcelona, Spain; VPerezSola@parcdesalutmar.cat (V.P.); MTorrens@parcdesalutmar.cat (M.T.); 5Institut de Neuropsiquiatria i Addiccions, Parc de Salut Mar, 08003 Barcelona, Spain; restevevila@gmail.com (R.E.); rsauras@parcdesalutmar.cat (R.S.); adrianaf2@hotmail.com (A.F.); CCastillo@parcdesalutmar.cat (C.C.); 6Imaging of Mood and Anxiety-Related Disorders (IMARD) group, Institut d’Investigacions Biomèdiques August Pi i Sunyer (IDIBAPS), 08036 Barcelona, Spain; quimradua@gmail.com (J.R.); al.solanes@gmail.com (A.S.); 7Early Psychosis: Interventions and Clinical-detection (EPIC) Laboratory, Department of Psychosis Studies, Institute of Psychiatry, Psychology & Neuroscience, King’s College London, London WC2R2LS, UK; 8Department of Clinical Neuroscience, Centre for Psychiatry Research, Karolinska Institutet, Solna, 17177 Stockholm, Sweden; 9Centro de Investigación Biomédica en Red de Salud Mental (CIBERSAM), 28029 Madrid, Spain; 10Department of Psychiatry, Universitat Autònoma de Barcelona, Bellaterra, 08193 Barcelona, Spain; 11RETICS-Redes Temáticas de Investigación Cooperativa en Salud en Trastornos Adictivos, 08003 Barcelona, Spain

**Keywords:** psychological trauma, posttraumatic stress disorder, substance use disorders, dual diagnosis, prevalence

## Abstract

Psychological trauma has been identified in substance use disorders (SUD) as a major etiological risk factor. However, detailed and systematic data about the prevalence and types of psychological trauma in dual disorders have been scarce to date. In this study, 150 inpatients were recruited and cross-sectionally screened on their substance use severity, psychological trauma symptoms, comorbidities, and clinical severity. One hundred patients fulfilled criteria for a dual disorder, while 50 patients were diagnosed with only SUD. Ninety-four percent of the whole sample suffered from at least one lifetime traumatic event. The prevalence rates of Posttraumatic Stress Disorder diagnosis for dual disorder and only SUD was around 20% in both groups; however, patients with dual disorder presented more adverse events, more childhood trauma, more dissociative symptoms, and a more severe clinical profile than patients with only SUD. Childhood maltreatment can also serve as a predictor for developing a dual disorder diagnosis and as a risk factor for developing a more complex and severe clinical profile. These data challenge our current clinical practice in the treatment of patients suffering from dual disorder or only SUD diagnosis and favor the incorporation of an additional trauma-focused therapy in this population. This may improve the prognosis and the course of the illness in these patients.

## 1. Introduction

Substance use disorder (SUD) is a psychiatric condition that affects judgement and alters cognitive functions, such as learning, memory, and impulse control [1]. SUD has a multifactorial etiology, resulting from the combination of different genetic [2] and environmental [3] factors. Moreover, like in many other psychiatric pathologies, a better or worse evolution is associated with multiple biological and socio-demographical variables, such as age of onset, access to drugs, social environment, race, and presence of external stressors [2,3,4]. Out of all these variables, psychological trauma has gained importance in clinical research studies, due to the strong negative impact it has on the onset, course, and prognosis of many psychiatric pathologies [5]. More than 70% of the adult population worldwide have experienced at least one psychological trauma event in their life, and 31% have suffered from four or more traumatic events. The majority recover from them without any external intervention [6]. However, those who do continue to experience symptoms related to psychological trauma are at risk of developing post-traumatic stress disorder (PTSD). Of note, in SUD samples, Posttraumatic Stress Disorder (PTSD) prevalence rates over the last 12 months vary from 15% to 41%, and lifetime rates vary from 26% to 52% [7]. These results are striking when compared to prevalence rates of current PTSD of 0.2–3.8% and lifetime prevalence of 1.3–12.3% in the general population [6]. There appears to be a greater vulnerability to develop, in general, somatic and/or psychiatric disorders when the traumatic event is experienced during childhood [8,9].

The relation between PTSD and SUD is a source of controversy. Briefly, the most accepted view is the “self-medication model” hypothesis, which means the traumatic event occurred prior to the substance use [10]. The “high-risk hypothesis” argues that substance abuse comes first, and this increases the probability of being exposed to more traumatic events, and in consequence to PTSD [11]. Thirdly, there is the “shared liability model”, which considers that both disorders develop simultaneously after the traumatic event due to a common biopsychosocial process [12,13,14]. Finally, the “susceptibility model” posits high anxiety and arousal as a consequence of chronic substance use, which in turn leads to a higher risk of PTSD [15]. Besides the lack of consensus between the different explanatory models, there does seem to be an agreement that SUD increases the severity of PTSD presentation, and PTSD seems to be an independent risk factor for an unfavorable outcome of SUD [4,16,17]. Specifically, patients with both disorders present a worse prognosis and evolution [4], a greater number of further comorbid somatic and psychiatric disorders [18], a higher number of detoxification treatment admissions and relapses [19], an earlier start to substance use [20], greater number of years of use [21], a poly-substance consumption pattern [22], a greater severity of PTSD symptoms [23], and a higher number, as well as a greater severity and intensity, of dissociative symptoms in those with poly-substance SUD [24]. In summary, patients with SUD and PTSD have a more severe and complex clinical profile [7].

In the last decade, epidemiological and clinical research has increasingly focused on the importance of detecting and treating the comorbidity of severe mental disorders and SUD, the so-called dual disorder [25]. Dual disorder patients, in comparison to only SUD patients, present a worse prognosis and a more severe and complex clinical profile, characterized by a greater number of further comorbid disorders [18], more associated medical and psychological problems, a higher number of detoxification treatment admissions [26], an earlier start to substance use [20], and a poly-substance consumption pattern [22]. However, despite the clear and strong association between PTSD and SUD, specific data about the prevalence and clinical characterization of psychological trauma in patients with dual disorders, and especially in comparison with SUD patients, are scarce so far.

Therefore, the present study aims to evaluate the prevalence and detailed characterization of traumatic psychological trauma and life events and PTSD and their relation to clinical variables in hospitalized dual disorder patients versus patients with only SUD. Our hypothesis was that patients with dual disorder would suffer from more adverse events, more childhood trauma, have more dissociative symptoms, and have a more severe clinical profile than patients with only SUD. Furthermore, we hypothesized that patients with dual disorder would have a higher prevalence of a PTSD diagnosis than patients with only SUD.

## 2. Materials and Methods

### 2.1. Participants

This multicenter collaborative study was conducted from 2017 to 2019 and involved the participation of three different dual pathology inpatient units (one at Hospital Benito Menni, Sant Boi and two at the Hospital Parc de Salut Mar) from the Barcelona catchment area, Spain. Our study sample is representative and a random representation of typical SUD patients with or without psychiatric comorbidity due to the nature of the two units. One is in the centre of Barcelona city (Hospital Parc de Salut Mar), in a mainly low and middle-class sociodemographic area and connected to the University, while the other one (Hospital Benito Menni, Sant Boi) is a community hospital, based in the outskirts of Barcelona in a mainly rural middle-class social catchment area, which widens the representativeness of our sample. The criteria for admission are the same in both centres, namely a clinical decompensation due to SUD and a comorbid psychiatric disorder. This means that there are no exclusion criteria (unless in the case of a severe somatic disorder) and patients cannot be rejected as long as they belong to the corresponding sector. Participants were selected for the following inclusion criteria: (1) admitted to an inpatient dual pathology unit; (2) aged between 18–65 years; (3) fulfilling Diagnostic and Statistical Manual of Mental Disorders, fifth edition, (DSM-5) criteria for SUD, based on a revised version for DSM-5 of the Spanish version of Psychiatric Research Interview for Substance and Mental Disorders (PRISM) [27] and (4) capable of speaking Spanish. Exclusion criteria were: (1) severe cognitive impairment; (2) organic brain syndrome; (3) suicidal thoughts; and (4) an acute psychotic state. Of the 517 eligible patients, 220 did not meet the inclusion criteria due to the following reasons: 199 were either in an acute psychotic state, clinically too unstable, or presented suicidal thoughts; 18 had marked cognitive impairment, and three did not speak and understand the Spanish language. Furthermore, 32 refused to participate, 62 requested early voluntary discharge and could not be evaluated, 36 were readmissions to the same unit, and 17 patients were not evaluated for other reasons. The final sample therefore consisted of 150 patients. Evaluation of the patients was carried out by clinical psychologists after an initial detoxification period during which clinical symptoms were stabilized. This was approximately two weeks from the day of admission.

The ethics committees of both hospitals approved the study (Benito Menni CASM: PR-2017-24 and Hospital Parc de Salut Mar: 2017/7650/I) and all participants signed written informed consent prior to enrollment. Participants did not receive any compensation for participation in the study.

### 2.2. Measures

Sociodemographic and some clinical variables were collected through an interview using a specific Case Report Form (CRF) designed for the study. The CRF collected data on sex, age, race, educational level, personal and family background, current pharmacological treatment, and drug use pattern. The latter included age of onset, quantity, frequency, and whether drug consumption started before or after experiencing a traumatic event.

Severity of addiction was assessed using the following scale:Severity of Dependence Scale (SDS) [28]: The SDS is a 5-item questionnaire which evaluates the degree of dependence on different types of drugs. Each item can be rated from 0 to 3, and higher scores mean greater dependence.

Psychological trauma symptoms were evaluated using the following tools:Global Assessment of Posttraumatic Stress Questionnaire (EGEP-5) [29]: The EGEP-5 is a 58-item questionnaire which evaluates current PTSD based on DSM-5 criteria. This questionnaire contains three different sections: (1) presence of traumatic events; (2) intensity of symptoms related to intrusion, avoidance, disturbances in cognition and mood, as well as activation and reactivity; (3) functionality in different areas of the person’s life.Childhood Trauma Questionnaire (CTQ) [30], Spanish validation [31]: The CTQ is a self-administered 28-item scale that measures five types of childhood maltreatment: emotional, physical, and sexual abuse, and emotional or physical neglect. A 5-point Likert scale is used for the responses, ranging from “Never True” to “Very Often True”.Dissociative Experiences Scale (DES) [32], Spanish validation [33]: This scale consists of a 28-item self-report questionnaire that measures different experiences related to dissociation. A total score higher than or equal to 30 indicates the presence of dissociation.The Holmes-Rahe Life Stress Inventory [34], Spanish validation [35]: This scale assesses the frequency of 43 common stressful life events over the last year. Scores below 150 reflect low levels of stress, scores between 150 and 299 represent a 50% risk of a stress-related illness in the near future, and scores above 300 represent an 80% risk of suffering from stress.

Comorbid disorders and clinical severity were assessed using the following instruments:
Dual Diagnosis Screening Interview (DDSI) [36]: The DDSI is a 63-item screening interview used to identify different psychiatric comorbidity in substance users, such as panic disorders, social phobia, agoraphobia, simple phobias, generalized anxiety, depression, dysthymia, mania, psychosis, attention deficit hyperactivity disorder, and PTSD.The diagnoses of any psychiatric comorbidity were confirmed using the corresponding module of the Spanish version of the Psychiatric Research Interview for Substance and Mental Disorders, revised for the DSM-5 (PRISM) [27].Hamilton Depression Rating Scale (HDRS) [37], Spanish validation [38]: This scale is a 17-item clinician-administered scale designed to identify depressive symptoms over the last week. Each item is scored on a 3- or 5-point scale, depending on the item, with a maximum score of 52. Scores are interpreted as follows: no depression (0–7), mild depression (8–16), moderate depression (17–23), and severe depression (≥ 24).Young Mania Rating Scale (YMRS) [39], Spanish validation [40]: The YMRS is an 11-item clinician-administered scale to evaluate hypomanic and manic symptoms over the last 48 h. Four items are scored from 0 to 8, while the remaining seven items are scored from 0 to 4. Higher scores mean greater severity.Brief Psychiatric Rating Scale (BPRS) [41], Spanish validation [42]: The BPRS is an 18-item clinician-administered scale which measures psychiatric symptoms, such as depression, anxiety, hallucinations, and unusual behavior. Each item is scored from 1 (not present) to 7 (extremely severe).

### 2.3. Statistical Analysis

For the purpose of this study, the sample of patients was divided into patients with a dual disorder diagnosis, and patients with only a SUD diagnosis. To describe the sample, we reported the means and standard deviations of the age, number of years of education, age of onset, number of substances used in the last year, and the scores of the clinical scales (HDRS, YMRS, BPRS, DES, SDS, CTQ). We reported the total number and percentage of the different groups of gender, nationality, relationship status, employment status, patient diagnosis, previous traumatic event, life events, axis 1 diagnosis, family background, and suicide attempts.

To investigate the clinical correlates of a dual diagnosis, we first assessed whether it was associated with increased depressive (HDRS), manic (YMRS), psychotic (BPRS), or dissociative symptoms (DES), whether childhood maltreatment (CTQ scores) was associated with having a dual diagnosis, and whether dual diagnosis was associated with the severity of the substance use disorders. Additionally, we repeated the same analysis to investigate the clinical correlates of gender differences.

To investigate the mental health consequences of childhood maltreatment in adults with substance use disorders, we assessed whether CTQ scores were associated with increased depressive (HDRS), manic (YMRS), or psychotic symptoms (BPRS), with an increased number of suicide attempts, and with the severity of the substance use disorders.

Finally, we conducted an analysis to study the association between the severity of the SUD and dissociative, intrusive, avoidance, and reactivity symptoms.

When the dependent variable was binary (e.g., having a dual diagnosis), we used logistic regressions covarying by age and sex. When the dependent variable was numeric (e.g., PRISM), we conducted standard regressions covarying by age and sex, but we found the statistical significance using the Freedman Lane permutation algorithm, which is very robust to violations of normality [https://doi.org/10.1016/j.neuroimage.2014.01.060]. This was necessary because most independent variables did not show a normal distribution and standard transformations to approximate normality were unsuccessful. All statistics were conducted in R Core Team (2020).

## 3. Results

### 3.1. Sample Characteristics

Sociodemographic data are shown in Table 1, and clinical data are presented in Table 2. 

Of the whole study sample, 100 patients fulfilled the diagnosis of a dual disorder, and 50 patients fulfilled only SUD diagnoses, according to DSM-5 criteria by PRISM. The most frequently used substances in the last month prior to the current admission included alcohol (*n* = 115), cocaine (*n* = 59), cannabis (*n* = 31), benzodiazepines (*n* = 20), opioids (*n* = 5), hallucinogens (*n* = 3), and amphetamines (*n* = 1). In the whole sample, 24 patients used one substance (16%), 67 patients used two substances (44.67%), and 58 patients three more substances (38.67%). The following types of medication were described in our sample: antipsychotics (*n* = 90), anticonvulsants (*n* = 58), antidepressants (*n* = 93), hypnotics (*n* = 46), and drugs for SUD (*n* = 17). Four patients (2.7%) did not take any medication.

With regard to traumatic experiences, in the whole sample, 141 patients (94%) reported at least one traumatic event in the EGEP-5 questionnaire. The death of a family member or close friend was the most prevalent (18%) event, followed by psychological abuse (15%), physical violence (13%), sexual violence (11%), severe accident (6%), and other adverse events (3%). Of those patients, 31 met criteria for current PTSD, following criteria of the EGEP-5. Dual disorder patients had a prevalence of PTSD diagnosis of 21%, and only SUD patients of 20%. In terms of adverse life events of the last year, all patients reported at least one of them. Finally, regarding childhood maltreatment, the results showed the subjects, on average, had experienced low-to-moderate levels of all types of child abuse and neglect in the CTQ, with both emotional abuse and neglect being the most frequent maltreatment reported by the patients (See Table 3). Minimization and denial in the CTQ were controlled for.

### 3.2. Clinical Differences between Patients with Dual Disorder Diagnosis and Only SUD Diagnosis

Patients with dual disorder diagnosis showed significantly higher scores in terms of depressive and psychotic symptoms in comparison with patients with a diagnosis of only SUD. Regarding the trauma variables, the results showed that dissociative symptoms and total CTQ score, as well as emotional abuse, sexual abuse, and physical neglect scores from the CTQ subscales, were also all statistically significantly higher in the dual disorder group than in the group with only SUD diagnosis. No significant differences were found between groups in terms of manic symptoms, nor in the severity of dependence on nicotine, alcohol, cannabis, and cocaine (see Table 3).

### 3.3. Association Between Clinical Symptoms and Childhood Maltreatment

In Table 4, the relationship between childhood maltreatment and depressive, manic, and psychotic symptoms, as well as the number of suicide attempts, can be seen. The HMDS and the BPRS scales showed an association with all variables of the CTQ, except for physical abuse. In contrast, the YMRS scale did not show any significant correlations with any variable of the CTQ. Finally, the number of suicide attempts showed only a significant correlation with emotional neglect.

### 3.4. Association Between Severity of Substance Dependence and Childhood Maltreatment and other Trauma-Related Variables

No significant correlations were found between the severity of dependence on nicotine, alcohol, cannabis, cocaine and childhood maltreatment or dissociative, intrusive, avoidance, and reactivity symptoms.

### 3.5. Childhood Maltreatment as Predictor of Dual Disorder Diagnosis

Using logistic regression, we found that childhood maltreatment can serve as a predictor for developing a dual disorder diagnosis (CTQ total score: z = 2.70; *p* = 0.006), with both emotional and sexual abuse being the most significant predictors (CTQ EA: z = 2.89; *p* = 0.003; CTQ SA: z = 2.36; *p* = 0.01).

### 3.6. Gender Differences of Clinical Variables

The sample consisted of 57 female and 93 male patients. We detected statistically higher scores for female patients in the total CTQ score and in the sexual abuse CTQ score in the total sample, when compared to male patients. We did not find any further sex-specific differences in clinical variables (see Table 5).

## 4. Discussion

To the best of our knowledge, this is one of the few studies to evaluate, in detail and systematically, the prevalence of psychological trauma and its association with clinical symptoms in a well-described and diagnosed sample of dual disorder patients versus only SUD patients. Additionally, we compared clinical variables in the whole sample dividing patients by gender.

The main analysis showed that two thirds of the whole sample fulfilled a diagnosis of dual disorder, all patients had an early onset of nicotine, alcohol, cannabis, and cocaine, and approximately one third had a positive family history of mood and SUD disorders. As expected, psychiatric symptoms were, in general, higher in the dual disorder sample than in the only SUD group. Regarding psychological trauma, the first overall result is that 94% of the whole sample suffered from at least one lifetime traumatic event, mainly related to deaths of relatives or friends and psychological, physical, and sexual abuse. These data are beyond the 70% of lifetime prevalence of one psychological trauma event found in the world-wide adult population [6], indicating that this population is vulnerable to suffering a greater number of negative life experiences than the general population [43]. Furthermore, our sample presented an overwhelming number of stressful life events in the 12 months prior to evaluation, supporting again the evidence of a high exposure to adverse events in both groups, which is similar to other psychiatric disorders, such as depression, bipolar disorder, or schizoaffective disorders [44,45,46,47].

Remarkably, of the whole sample, 20.67% met criteria for current PTSD. Dual disorder patients had a prevalence rate of 21%, and only SUD patients of 20%. These data are higher than the prevalence rates of current PTSD of 0.2–3.8% in the general population [6], and within the range of prior data for only SUD patients with current PTSD, which range from 15% to 41% [7]. Despite our results being consistent with prior studies, which have shown that an important proportion of patients with only SUD have suffered from traumatic events and present marked PTSD symptomatology, our findings do not support our previous hypothesis that dual disorder patients will show a higher PTSD prevalence rate than patients with only SUD. However, patients with dual disorder presented more adverse events, more childhood trauma, more dissociative symptoms, and a more severe clinical profile than patients with only SUD. Specifically, total scores and scores of emotional and sexual abuse and physical neglect in the CTQ were higher in the dual disorder group when compared with the only SUD group. These results support previous research that have found that early traumatic experiences, especially childhood maltreatment, are not only a risk factor for developing several mental health problems in adulthood, including SUD, psychosis, depression, or bipolar disorder [48,49,50], among others, but that they represent also a risk factor for developing a more severe clinical presentation in dual disorder patients [51]. As a matter of fact, we also found that childhood maltreatment can serve as a predictor for developing a dual disorder diagnosis, with both emotional and sexual abuse being the most significant predictors. These data are similar to the results found by Fetzner et al. (2011), which suggested that childhood maltreatment is a predictor for the course of SUD, even in the absence of comorbid PTSD [52]. However, interestingly, while both groups of patients scored more highly across subtypes in the CTQ than the general population, their scores were in the low–moderate range, which is not in line with prior literature in only SUD populations (e.g., [53]). One possible explanation could be the predominance of male patients in our sample, as the prevalence of childhood adversity is higher in women with SUD than in men [54,55]. In fact, we detected that women specifically showed higher scores in the total score and sexual abuse score in the CTQ as compared to men. We also did not include in our analysis patients with an acute psychotic episode. This might have influenced these results, as adversity in childhood is an etiological risk factor for developing psychosis and might therefore be more prevalent in dual disorder than in patients with mood or anxiety disorders [56].

The dual disorder sample also presented higher scores in depressive and psychotic symptoms compared to patients with only SUD. This can be expected, due to the nature of a comorbid psychiatric disorder, but it might also indicate a more severe and complex clinical profile in a dual disorder, as suggested in a study by Sells et al., 2016 [18]. Moreover, dual disorder patients also showed higher scores in dissociation and depersonalization, but not in amnesia when compared to the only SUD group, which also underlines a more complex clinical picture and a higher trauma load in this sample. Interestingly, dissociative scores of our sample were, in general, also in the lower range. These data are consistent with a recent meta-analysis which assessed dissociation in several mental disorders and reported that the largest dissociation scores were found for dissociative disorders, followed by PTSD, borderline personality disorder, and conversion disorder, and the lower range of scores included substance-related and addictive disorders, schizophrenia, anxiety disorder, and affective disorders, amongst others [57]. Regarding the severity of dependence on nicotine, alcohol, cannabis, and cocaine, both groups showed a similar pattern of consumption, which is surprising due to prior evidence that dual disorder patients have an earlier start to substance use [20], and a poly-substance consumption pattern [22].

With respect to the relationship between psychiatric symptoms and the number of suicide attempts and childhood maltreatment, we found that the HDRS and the BPRS scales showed an association with all variables of the CTQ, except for physical abuse. This points towards the influence of childhood maltreatment as a risk factor for developing a variety of psychiatric symptoms in adult life, as also suggested in prior literature [58,59,60]. In contrast, the YMRS scale interestingly did not show any significant correlations with any variable of the CTQ. This is of interest, as, for example, the Kessler et al. study from 1995 detected, in a large PTSD sample, a high risk, especially in men, of developing manic episodes in the long-term. However, psychopathological scores were, in general, low in our sample, as patients were evaluated once their clinical symptoms were stabilized, meaning these results must be interpreted with caution.

Finally, the number of suicide attempts showed a significant correlation with emotional neglect in the CTQ. This finding is in line with a recent meta-analysis, which showed a two- to three-fold increased risk for suicide attempts and suicidal ideation in adults who experienced childhood adversities compared with adults who have not experienced maltreatment during childhood [61]. Therefore, these data support previous studies which suggest that childhood maltreatment can aggravate the clinical symptoms of existing psychopathologies [49,50,51].

Our work includes various limitations. One is the cross-sectional nature of our study and the lack of a longer stabilization phase. Furthermore, we did not clearly define a stabilization phase using a determined score range in psychopathological scales during a longer period of time to evaluate our patients. This was not planned as such due to a possible low adherence and short admission duration in this clinically complex population. However, both aspects might have possibly influenced our results. There was also a slight predominance of male patients, meaning results cannot be completely generalized to female patients. Furthermore, dual disorder patients with acute psychotic states were excluded, meaning the dual disorder sample consisted mainly of comorbid mood and anxiety disorders. This limits its representativeness across the wide psychiatric diagnostic spectrum, including schizoaffective disorder and schizophrenia. The main reason for this was that we considered that psychotic states needed more time for stabilization beyond the median duration of a stay of 20 days in both units. We considered that psychopathological instability would influence the evaluation. We also excluded a smaller number of patients with cognitive impairment, as this was highly likely to have also influenced the quality of our evaluation. Patients with suicidal thoughts were also not included, as the evaluation of traumatic events might have worsened suicidal thoughts when no trauma-focused therapy was offered. Our hypothesis is that, if these exclusion criteria did influence results in any way, it would have led to a lower estimation of prevalence rates. There is, for example, compelling evidence that one major etiological factor of psychosis is childhood trauma [56] Broadening the diagnoses in future studies might overcome this limitation.

Strengths of our work include the systematic investigation of psychological trauma and life events in a large sample of patients using established and validated scales in Spanish. Furthermore, we used a gold-standard clinical structured interview, the PRISM, following DSM-5 criteria, to establish SUD or dual disorder diagnosis together with the DDSI. Therefore, our level of confidence in diagnosis, and especially in the prevalence estimate is high.

## 5. Conclusions

In conclusion, we found a high rate of traumatization in the form of negative life events or PTSD throughout the sample, with some types of childhood maltreatment as a predictor of a dual diagnosis and as a risk factor to develop a more complex and severe clinical profile. The prevalence rates of PTSD in dual disorder and only SUD patients were around 20%, which means that one in five dual disorder patients actually have a triple diagnosis. Our data therefore challenge our current clinical practice in the treatment of patients suffering from dual or only SUD diagnosis, and favor the incorporation of an additional trauma-focused strategy in this population, such as trauma-focused psychological interventions, namely cognitive behavioural therapy [62] or Eye Movement Desensitization Reprocessing (EMDR) therapy [63,64]. This may improve the prognosis of the often. complex course of illness in individuals suffering from dual disorder or only SUD.

## Figures and Tables

**Table 1 jcm-09-02553-t001:** Sociodemographic characteristics of the sample. Data are presented as mean (SD) or number (%).

	Total Sample (*n* = 150)	Dual Disorders (*n* = 100)	Only SUD (*n* = 50)
Age	44 (10)	44.4 (10)	43.2 (10)
Gender Female Male	57 (38%) 93 (62%)	44 (44%) 66 (66%)	13 (26%) 37 (74%)
Nationality Spanish Latin Moroccan Other	134 (89.3%) 8 (5.3%) 3 (2%) 5 (3.3%)	86 (86%) 8 (8%) 3 (3%) 3 (3%)	48 (96%) 0 (0%) 0 (0%) 2 (4%)
Education (years of studies)	11.2 (3.4)	11.2 (3.5)	11.2 (3.2)
Relationship status Single Married Separate/divorce Widowed	58 (38.7%) 39 (26%) 49 (32.6%) 4 (2.7%)	35 (35%) 28 (28%) 35 (35%) 2 (2%)	23 (46%) 11 (22%) 14 (28%) 2 (4%)
Employment status Student Full time employment Part-time employment Sick leave Unemployed Work incapacity by mental health problems Work incapacity by other reasons	1 (0.07%) 11 (7.3%) 1 (0.07%) 55 (36.7%) 46 (30.7%) 26 (17.3%) 10 (6.6%)	1 (1%) 3 (3%) 1 (1%) 37 (37%) 28 (28%) 24 (24%) 6 (6%)	0 (0%) 8 (16%) 0 (0%) 18 (36%) 18 (36%) 2 (4%) 4 (8%)

**Table 2 jcm-09-02553-t002:** Clinical characteristics of the sample. Data are presented as mean (SD) and/or number (%).

	Total Sample (*n* = 150)	Dual Disorder (*n* = 100)	Only SUD (*n* = 50)
Age of onset Nicotine Alcohol Cannabis Cocaine Heroin Stimulants Sedatives	15.3 (3.7) *n* = 141 * 15.1 (3.8) *n* = 143 * 16.8 (5) *n* = 80 * 20.9 (7.8) *n* = 97 * 29.1 (8.9) *n* = 11 * 19 (5.3) *n* = 14 * 34.4 (10.1) *n* = 24 *	15 (4) *n* = 94 * 15.4 (4.1) *n* = 95 * 17.2 (5.7) *n* = 52 * 21.2 (7.6) *n* = 66 * 29 (9.3) *n* = 10 * 20.3 (6.9) *n* = 7 * 34.6 (10.9) *n* = 19 *	15.7 (3) *n* = 47 * 14.7 (3.3) *n* = 48 * 16.1 (3.2) *n* = 28 * 20.3 (8.2) *n* = 31 * 30 (-) *n* = 1 * 17.7 (2.9) *n* = 7 * 33.6 (7.4) *n*=5 *
Number of drugs in the last year	2.28 (0.93)	2.29 (0.93)	2.26 (0.94)
Previous traumatic event No Yes PTSD diagnosis Non-PTSD diagnosis Live events (last 12 months) From 1 to 5 events From 6 to 10 events From 11 to 15 events From 16 to 20 events From 21 to 25 events >26 events	9 (6%) 141 (94%) 31 (20.67%) 110 (73.33%) 42 (28%) 62 (41.3%) 33 (22%) 8 (5.3%) 4 (2.7%) 1 (0.7%)	3 (3%) 97 (97%) 21 (21%) 76 (76%) 29 (29%) 40 (40%) 22 (22%) 6 (6%) 2 (2%) 1 (1%)	6 (12%) 44 (88%) 10 (20%) 34 (68%) 13 (26%) 22 (44%) 11 (22%) 2 (4%) 2 (4%) 0 (0%)
Comorbid diagnosis axis 1 Mood disorders Anxiety disorders Psychotic disorders Induced psychotic or mood disorders Eating disorders	37 (24.7%) 4 (2.7%) 7 (4.7%) 18 (12%) 1 (0.7%)	37 (37%) 4 (4%) 7 (7%) 2 (2%) 1 (1%)	0 (0%) 0 (0%) 0 (0%) 16 (32%) 0 (0%)
Family history Father None SUD Mood disorders SUD + other Mother None SUD Mood disorders SUD + other Sibling None SUD Mood disorders SUD + other	107 (71.3%) 34 (22.7%) 4 (2.7%) 5 (3.3%) 111 (74%) 6 (4%) 25 (16.7%) 6 (4%) 97 (64.7%) 33 (22%) 12 (8%) 6 (4%)	72 (72%) 21 (21%) 3 (3%) 4 (4%) 72 (72%) 3 (3%) 17 (17%) 6 (6%) 58 (58%) 24 (24%) 11 (11%) 6 (6%)	35 (70%) 13 (26%) 1 (2%) 1 (2%) 39 (78%) 3 (6%) 8 (16%) 0 (0%) 39 (78%) 9 (18%) 1 (2%) 0 (0%)
Suicide attempts None One Two Three or more	72 (48%) 33 (22%) 20 (13.3%) 43 (28.7%)	38 (38%) 23 (23%) 15 (15%) 37 (37%)	34 (68%) 10 (20%) 5 (10%) 6 (12%)

SUD: Substance Use Disorder; PTSD: Posttraumatic Stress Disorder; * Number of patients who consume the substance.

**Table 3 jcm-09-02553-t003:** Clinical differences between patients with dual disorder diagnosis and only Substance Use Disorder (SUD) diagnosis. Data are presented as mean (SD).

	All Sample (*n* = 150)	Dual Disorder (*n* = 100)	Only SUD (*n* = 50)	*p*-Value (a)
HDRS	7 (5.2)	7.7 (5.4)	5.6 (5.4)	**0.04**
YMRS	1.3 (2.7)	1.5 (2.9)	0.6 (2.9)	0.109
BPRS	24.3 (5)	25.1 (5.2)	22.8 (5.2)	**0.005**
DES Total score	10.8 (9.4)	11.9 (10.3)	8.4 (10.3)	**0.02**
Amnesia	6.9 (8.2)	7.4 (9.1)	5.8 (9.1)	0.25
Dissociation	16.1 (12.7)	17.7 (13.5)	12.9 (13.5)	**0.014**
Depersonalization	6.4 (8.6)	7.6 (9.8)	3.9 (9.8)	**0.009**
SDS-Nicotine SDS-Alcohol	8.6 (3.8) 9.5 (3.4)	8.5 (4) 9.6 (3.3)	8.9 (4) 9.4 (3.3)	0.67 0.81
SDS-Cocaine	9.8 (4.1)	9.7 (4.3)	9.8 (4.3)	0.64
SDS-Cannabis	8.4 (4.2)	8.6 (4.6)	7.7 (4.6)	0.42
CTQ Total score	44.4 (17)	47.3 (18.7)	38.8 (18.7)	**0.003**
Emotional abuse	10.4 (5.3)	11.4 (5.5)	8.6 (5.5)	**0.001**
Physical abuse	7.6 (4.1)	8.1 (4.5)	6.7 (4.5)	0.067
Sexual abuse	6.8 (3.9)	7.5 (4.6)	5.5 (4.6)	**0.009**
Emotional neglect	11.7 (5)	12.2 (5.2)	10.8 (5.2)	0.12
Physical neglect PTSD	7.7 (3.3) 31 (20.7%)	8.1 (3.5) 21 (21%)	7 (3.5) 10 (20%)	**0.042** 0.82

HDRS: Hamilton Depression Rating Scale; YMRS: Young Mania Rating Scale; BPRS: Brief Psychiatric Rating Scale; DES: Dissociative Experiences Scale; SDS: Severity of Dependence Scale; CTQ: Childhood Trauma Questionnaire; (a) *p*-value derived from the comparisons between individuals with dual diagnosis and individuals with no dual diagnosis via logistic regressions or Freedman-Lane permutation tests covarying for age and sex. Numbers in bold are statistically significant.

**Table 4 jcm-09-02553-t004:** Freedman Lane analysis to evaluate the relation between the childhood trauma questionnaire scores and clinical variables from the HDRS, YMRS, BPRS, and the suicide attempts.

CTQ	HDRS	YMRS	BPRS	SA
Total score	R = 0.28, t = 3.54, ***p* ≤ 0.01**	R = 0.04, t = 0.53, *p* = 0.53	R = 0.23, t = 2.91, ***p* = 0.01**	R = 0.16, t = 2.00, *p* = 0.06
Emotional abuse	R = 0.17, t = 2.06, ***p* = 0.03**	R = 0.006, t = 0.08, *p* = 0.91	R = 0.17, t = 2.11, ***p* = 0.03**	R = 0.14, t = 1.75, *p* = 0.09
Physical abuse	R = 0.16, t = 2.02, *p* = 0.06	R = 0.08, t = 0.95, *p* = 0.33	R = 0.15, t = 1.88, *p* = 0.06	R = 0.07, t = 0.88, *p* = 0.2
Sexual abuse	R = 0.23, t = 2.86, ***p* = 0.01**	R = 0.14, t = 1.71, *p* = 0.12	R = 0.18, t = 2.17, ***p* = 0.03**	R = 0.11, t = 1.32, *p* = 0.24
Emotional neglect	R = 0.28, t = 3.61, ***p* ≤ 0.01**	R = −0.06, t = 0.79, *p* = 0.39	R = 0.21, t = 2.55, ***p* = 0.01**	R = 0.16, t = 2.02, ***p* = 0.04**
Physical neglect	R = 0.24, t = 3.04, ***p* = 0.01**	R = 0.05, t = 0.65, *p* = 0.49	R = 0.19, t = 2.32, ***p* = 0.02**	R = 0.16, t = 2.00, *p* = 0.13

HDRS: Hamilton Depression Rating Scale; YMRS: Young Mania Rating Scale; BPRS: Brief Psychiatric Rating Scale; SA: suicide attempts; R converted from the t obtained in the Freedman-Lane linear model. Numbers in bold are statistically significant.

**Table 5 jcm-09-02553-t005:** Clinical differences by gender. Data are presented as mean (SD).

	All Sample (*n* = 150)	Women (*n* = 57)	Men (*n* = 93)	*p*-Value (a)
HDRS YMRS BPRS DES Total score Amnesia Dissociation Depersonalization	7.03 (5.25) 1.28 (2.67) 24.33 (4.97) 10.76 (9.37) 6.88 (8.24) 16.12 (12.72) 6.37 (8.63)	8.05 (5.18) 1.3 (2.76) 24.11 (4.02) 10.87 (10.43) 7 (9.34) 15.72 (13.23) 6.75 (9.74)	6.41 (5.22) 1.27 (2.63) 24.46 (5.5) 10.7 (8.71) 6.81 (7.55) 16.37 (12.47) 6.13 (7.93)	0.086 0.884 0.604 0.822 0.872 0.856 0.588
SDS-Alcohol	9.5 (3.4)	9.82 (2.97)	9.31 (3.64)	0.438
SDS-Cocaine	9.76 (4.14)	9.45 (4.41)	9.89 (4.06)	0.788
SDS-Cannabis	8.37 (4.22)	7.83 (4.34)	8.65 (4.23)	0.67
SDS-Nicotine	8.63 (3.82)	8.23 (4.43)	8.86 (3.45)	0.37
CTQ Total score	44.44 (16.9)	48.05 (20.64)	42.23 (13.79)	**0.034**
Emotional abuse	10.45 (5.26)	11.28 (5.89)	9.94 (4.79)	0.078
Physical abuse	7.64 (4.12)	8.3 (5.11)	7.24 (3.35)	0.124
Sexual abuse	6.82 (3.95)	8.65 (5.46)	5.7 (1.94)	**<0.001**
Emotional neglect	11.73 (5.05)	12.14 (5.2)	11.48 (4.96)	0.448
Physical neglect	7.75 (3.32)	7.68 (3.69)	7.78 (3.1)	0.946

HDRS: Hamilton Depression Rating Scale; YMRS: Young Mania Rating Scale; BPRS: Brief Psychiatric Rating Scale; DES: Dissociative Experiences Scale; SDS: Severity of Dependence Scale; CTQ: Childhood Trauma Questionnaire; (a) *p*-value derived from the comparisons between individuals with PTSD diagnosis and non-PTSD diagnosis via logistic regressions or Freedman-Lane permutation tests covarying for age and sex. Numbers in bold are statistically significant.

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
