# Peer review of "Traumatic Events in Dual Disorders: Prevalence and Clinical Characteristics"

_jcm, 2020, doi:10.3390/jcm9082553_

Round 1

Reviewer 1 Report

It is well written and directly adresses the research question of interest while acknowledging its limitations. 

The comments below are for further improvement or left me with unanswered questions as a reader (in no particular order).

  • The representative nature of the patients with SUD admitted to these specific centres where the sample was drawn (described as dual pathology inpatient units). Is there any specific referral criteria diverting a subgroup of people with SUDs to such units or they are a random representation of people with substance use disorders in the Barcelona catchment area.
  • While it is mentioned in a way in the limitation, amongst the 297 patients meeting the inclusion criteria (517 - 220), at least a third were not assessed for various reasons reducing the sample to 150 patients. How could this have affected the conclusions? perhaps this could be further elaborated more to strengthen conclusions.

Minor cosmetics: 

  • Table 2 has a typo (heroin not heroine)
  • Table 4 was hard to read, perhaps a space between subscale measures would be useful

Aside that it is a very welcomed addition to the scientific literature. 

Author Response

We thank the reviewer for the overall positive comments and further constructive suggestions!

  1. The reviewer wonders about whether admission to the two dual disorder units is representative: this is an important issue which needs to be better explained. We believe that our study sample is representative and a random representation due to the nature of the two units. One is linked to the University, in the centre of Barcelona city, in a mainly low and middle-class sociodemographic area, while the other one is a community hospital based more in the outskirts of Barcelona in a mainly rural middle-class social catchment area. To our mind, the city (and University) and rural characteristics of the two centres widen the representativeness of our sample. The criteria for admission are the same in both centres, namely a clinical decompensation due to SUD and a comorbid psychiatric disorder. This means that there are no exclusion criteria (except of a clear somatic disorder) and patients cannot be rejected as long as they belong to the corresponding sector. This has been added now to the method section.
  2. Of the 517 patients admitted to the units over the study period, 220 patients did not meet inclusion criteria. A further 147 patients did refuse to participate or asked for voluntary dismission. 150 patients were finally evaluated. The reviewer wonders whether or not that might have influenced results. As stated in the limitation section, we did exclude patients with acute psychotic episodes, suicidal thoughts or with cognitive impairment. The main reason for this was that we considered that psychotic states needed more time for stabilization beyond the median duration of stay of 20 days in both units. We considered that psychopathological instability would influence the quality of the evaluation. Furthermore, cognitive impairment was highly likely to have also influenced the evaluation. We also excluded patients with suicidal thoughts as we considered that the evaluation of traumatic events might worsen suicidal thoughts when there is no trauma-focused therapy offered. Our hypothesis is that, if these exclusion criteria did influence results in any way, it would have led to a lower estimation of prevalence rates. There is, for example, compelling evidence that one major etiological factor of psychosis is childhood trauma (Varese et al, 2012). Our hypothesis is that, by including psychotic disorders, we possibly would have detected a higher prevalence of childhood trauma. Broadening the diagnoses in future studies might overcome this limitation. We added further information on this in the limitation and discussion section. 
  3. The type and minor cosmetics of table 4 have been revised and included in the new version.   

Reviewer 2 Report

Overall, the article is well-written. The introduction provides sufficient background information, the methods are adequately described, and the results are clear and concise. I would suggest minor English review first. 

Author Response

We thank the reviewer for the overall positive comments. 

We have asked an English native speaker to revise the document to improve minor spelling and grammatical errors.